# A graph-based visualization for monitoring of high-performance computing system

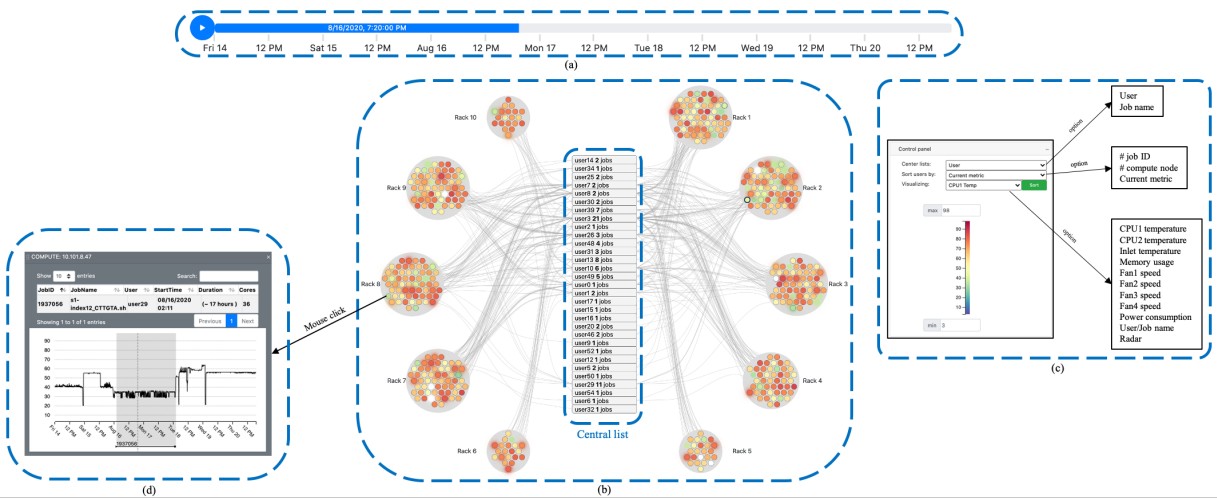

Figure 1: The main components of JobViewer: (a) timeline, (b) main visualization, (c) control panel, (d) table of jobs' information.

## ABSTRACT

Visualization aims to strengthen data exploration and analysis, especially for complex and high dimensional data. High-performance computing (HPC) systems are typically large and complicated instruments that generate massive performance and operation data. Monitoring the performance and operation of HPC systems is a daunting task for HPC admins and researchers due to their dynamic natures. This work proposes a visual design using the idea of the bipartite graph to visualize the monitoring the structural and metrics data of HPC clusters. We build a web-based prototype, called JobViewer, that integrates advanced methods in visualization and human-computer interaction (HCI) to demonstrate the benefits of visualization in real-time monitoring an HPC at a university. We also show real use cases and a user study to validate the efficiency and highlight the drawbacks of the current approach.

**Index Terms:** Human-centered computing—Visualization—Visualization application domains—Visual analytics;

## 1 INTRODUCTION

High-performance computing (HPC) systems can provide powerful computing resources for many scientific fields, such as quantum chemistry, bioinformatics, high energy physics, and many others. TheESE typically complex research instruments ranging from hundreds to thorusand of computing nodes require substantial efforts in monitoring to ensure the trade off of cost versus performance. One approach that can strengthen monitoring efficiency is to apply visualization and human-computer interaction (HCI) to the operational data. The HCI theory relies on cognitive principles to design the user interface and interactive activities [11], so it allows HPC administrators to gain necessary information quickly and intuitively. This work aims to apply the advantages of HCI to monitoring data

at an HPC center [22] to demonstrate the benefits of visualization in monitoring activities.

Monitoring tasks vary significantly for different administrators and their purposes, but the usual activities are to look at the current states of the system or analyze the historical data for the overview of long-term trends [1]. The administrators often consider the compute node health because they can know what happens with the system by analyzing memory usage, CPU usage, temperature, etc... They may also put their attention into job state information to assess the operation of the system. It is interesting to analyze both users' jobs information and the compute node health because the combined view can help the administrators understand how these jobs utilize the set of resources in the system and know the relations between users' activities and the system's state. Moreover, the knowledge about regular users or jobs' behaviors can encourage the administrators to improve their HPC performance. Therefore, we target the users' jobs and the compute node health for developing an interactive web-based prototype, called JobViewer, to provide a novel monitoring aspect for an HPC system.

The main contribution of this work is three folds.

- We apply HCI's advantages to visualize users' jobs and node health monitoring of an HPC system by building a web-based prototype, namely JobViewer, for this purpose.

- We illustrate the benefits of monitoring both the above aspects in their relations with some real use cases.

- We carry out a user study to verify whether the approach and designs are suitable for practical uses.

This paper's structure is as the following. The next section covers some related works, and then, section 3 will consider the designs of the proposed web-based prototype. Section 4 discuss real use cases of the visualizations, while section 5 mentions user study for the approach. Finally, section 6 discuss all results of this work and section 7 summarizes the work.

## 2 RELATED WORKS

### 2.1 HPC performance monitoring

HPC monitoring is not a new problem, so there are several well-known performance analysis tools, both commercial and open-source ones. Ganglia is an open-source distributed monitoring system for clusters and grids. Ganglia's strength is the scalability, with some measurements showing that Ganglia can scale on clusters of up to 2000 nodes and federations of up to 42 sites [14]. This tool uses RRDtool [19] to store and visualize time series data. Nagios [8] is another tool that many organizations utilize. It can be suitable for monitoring a variety of servers and operating systems with industrial standards. The tool has two versions: one commercial (Nagios XI) and open-source (Nagios Core). The commercial version has web interface and performance graphing [3]. However, there are some issues with traditional Nagios including:

- Nagios requires human intervention for the definition and maintenance of remote hosts configurations in Nagios Core.

- Nagios requires Nagios Remote Plugin Executor on Nagios Server and each monitored remote host.

- Nagios mandates Nagios Service Check Acceptor (NSCA) on each monitored remote host.

- Nagios also requires to check specific agents (e.g. SNMP) on each monitored remote host.

Besides, CHReME [16] provides a web-based interface for monitoring HPC resources that took non-expert away from conventional command lines. This tool, however, focuses on basic tasks which can also be found on Nagios engine. Splunk [5] is another software platform for mining and investigating log data for system analysts. Its most significant advantages are the capability to work with multiple data types (e.g., csv, json or other formats) in real-time. It has been used and shown consistent performance in the study [21, 26]. However, Greenberg and Debardeleben [12] pointed out that Splunk was not feasible for searching a vast amount of data generated every day (e.g., hundreds of gigabytes of data) due to slow performance. Grafana [9] provides a vibrant interactive visualization dashboard which enables users to view metrics via a set of widgets (e.g., text, table, temporal data). Grafana defines a place holder (i.e., arrays) that automatically generates widgets based on its values. This also a limitation of Grafana: customized visualizations (such as parallel coordinates [20] and scatterplot matrices [24] for analyzing high-dimensional data) are not supported. This visualization package has been used in [4, 12] due to its multiple data stores features. Windows Azure Diagnostic or Amazon cloud watch [13], are also common tools for performance monitoring purposes. The survey of these tools [3] can give more details of interest.

### 2.2 Time Series Visualizations

One crucial factor that we need to consider if we want to do visualizations is the data structure. This work investigates a high dimensional temporal dataset with four dimensions: 1) User and job, 2) Compute node, 3) Health metrics, and 4) Time. In other words, we consider the data of 467 compute nodes at an HPC system [22]. Each compute node has nine health metrics, including two CPU temperatures, inlet temperature, four fans' speed, memory usage, and power consumption. Each metric of a compute node is recorded every 5 minutes to form a time series. Moreover, users utilize the compute nodes to run their jobs. If we ignore the user and job, this data becomes the panel data, and there are various ways to visualize it. One example is TimeSeer [6], which transforms the panel data into time series of Scagnostics. The Scagnostics are measures for scoring point distribution in scatterplots [25]. The main idea of TimeSeer is to use these measures as a sign to quickly identify time

steps with rare events. Another method is the use of connected scatterplots for displaying the dynamic evolution of pairwise relations between variables in the data [18]. Besides, parallel coordinates can also be extended for the panel data [2, 7, 23]. However, these common projections' extensions cannot visualize the relationships between users' jobs with the compute nodes. It is the reason why we propose a novel design of visualization for the dataset with four dimensions, and the detailed discussions are in the next section.

## 3 DESIGN DESCRIPTIONS

Based on our weekly discussions with the domain experts, the HPC visualization requirements are expanded on the following dimensions: HPC spatial layout, temporal domain, resource allocations and usages, and system health metrics such as CPU temperature, memory usage, and power consumption. We therefore focus the following design goals on: (**D1**) Provides spatial and temporal overview across hosts and racks, (**D2**) Provides the holistics overview of the system on a health metrics at a selected timestamp. (**D3**) Highlights the correlation of system health services and resource allocation information within a single view, and (**D4**) Allows system administrators to drill down a particular user/job/compute to investigate the raw time series data for system debugging purposes. Figure 1 depicts four main components of the JobViewer, including the timeline, main visualization, control panel, and the job table. Let first consider the timeline in Figure 1(a) We use animation to illustrate the temporal flow of the dataset. Animation has positive impacts on cognitive aspects such as interpreting, comparing, or focusing [17]. Although this method cannot grasp the whole temporal information, it is convenient for both uses: analyzing historical data and visualizing the system lively. The timeline has another benefit in quickly investigating time steps of interest.

Figure 1(b) shows the main visualization at a particular time step. The design bases on the idea of the bipartite graph with two disjoint sets of vertices. One set contains users, and another consists of the compute nodes. The link between a user with a compute node implies that the node is running at least one job of the user (Design goal **D3**). We design this graph with all users in the central list, and all compute nodes surrounding it. The compute nodes are divided into ten racks as their actual spatial locations. A benefit of graph-based visualization is that it is easy to highlight the link between a user and a compute node. For instance, we implement the mouse over the user or the compute node to highlight the corresponding vertex's links. The graph-based design also allows us to apply a simple visual method for illustrating the compute nodes' health metrics. JobViewer uses color to display the value of a chosen metric. The map from color to value is depicted by the color scale on the control panel tab, as can be seen in Figure 1(c). We use these simple visual presentations to display all four dimensions of the dataset mentioned in the previous section.

Besides the above designs, we also implement others to give related information and improve the cognitions. It is easy to recognize a new user appearing on the central list; however, if a current user, who has some jobs running somewhere in the system, run a new job, it is difficult to identify. If this case happens, we highlight the user at the corresponding time step by its outline and the color of links to compute nodes allocated the new jobs. About the compute nodes, one may wonder how many jobs a compute node is running. We visualize the number of jobs running on a compute node by the thick of its outline. Additionally, if the chosen metric's value on a compute node varies significantly over two consecutive time steps, we use the blur effect to highlight the sudden change.

Figure 1(c) shows the control panel and all options of the drop-down menus. There are two options for displaying on the central list: one offers the user, and another gives the job name. The next function is ranking that sorts all users on the list by a chosen option. Three options for the ranking are the number of jobs, the number of

compute nodes utilized by the user or job, and the selected health metric. We can also select one of the nine metrics to visualize by the compute nodes' color in the visualizing tab. Two more options beyond health metrics in the visualizing tab are user/job name and radar view [15]. If we select the user/job option, all compute nodes are colored according to their users/jobs. If we select the radar view, JobViewer visualizes every compute node by a radar chart representing all its health metrics.

However, if we observe the compute nodes by their radar chart, it is difficult to recognize the shapes because their size is relatively small. We found that the use of color is more effective for cognitive activities. It is the reason why we apply some clustering algorithm to cluster the compute nodes based on their health metrics. Then, we color each cluster a different color. Every radar chart representing a compute node has the color of its group. This method also improves the analyzing process because it reduces 467 compute nodes to a much smaller number of patterns of health states. We can quickly gain characteristics of the system states or detect strange behaviors of some compute nodes. Two clustering algorithms integrated into JobViewer are k-means and leader algorithm [10].

Another interaction with the main visualization is the mouse click to show the corresponding table of jobs' information, as seen in figure 1.d. On the table, we can find all information related to the jobs, such as the job's identity, job name, users, number of cores the job is using, etc. (Design goal **D4**). Moreover, we also display the time series of the selected health metric on the clicked compute node, and we highlight the period when the job runs on that compute node by the grey area. This visualization helps us understand what happens with compute nodes when particular users are using them. As we will show in the next section, this feature may give information about relations between jobs or users with compute nodes' health states.

To sum up, JobViewer designs the visualization using the idea of a bipartite graph. It also integrates some simple visual methods and clustering algorithms to improve cognitions for the analyzing process. The next sections demonstrate how we can use JobViewer in monitoring an HPC system.

## 4 USE CASES

### 4.1 Job allocation

The first use case focuses on how JobViewer provides information about job allocations. Figure 2.a shows a snapshot of the main visualization on 08/14/2020 at 5:50 PM. The color distinguishes between different users, along with their related compute nodes. If a compute node runs several jobs of multiple users, it has all corresponding colors. If a compute node is white, no user's job is running on it. At 5:50 PM, there are nine white compute nodes that locate in six different racks. Ten minutes later, *user0*'s job starts, as highlighted by the black outline and links in figure 2.b. It takes 1080 cores, or 30 compute nodes (each compute node has 36 cores). The system allocates seven out of nine white compute nodes to this job, and there are still two white compute nodes at 6:00 PM. One is on rack 2, and another is on rack 9. Figure 2.c highlights all 30 compute nodes running the *user0*'s job. From these 30 compute nodes, 18 ones run two jobs, and 12 others run only one job. We have checked and found that most of the 18 compute nodes' former jobs consume all 36 cores at 5:55 PM. It means some of the compute nodes utilize up to 72 cores, including virtual cores, at 6:00 PM. These figures show information about job scheduling. Although there are two unused compute nodes, and the job requires so many cores to run, the system reuses the compute nodes running another job and does not allocate the two unused ones to the job. This use case is an example that can illustrate how efficient JobViewer can help HPC administrators to monitor job scheduling.

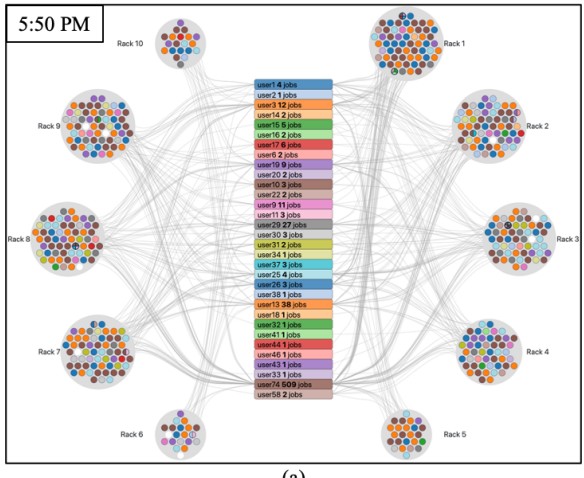

(a)

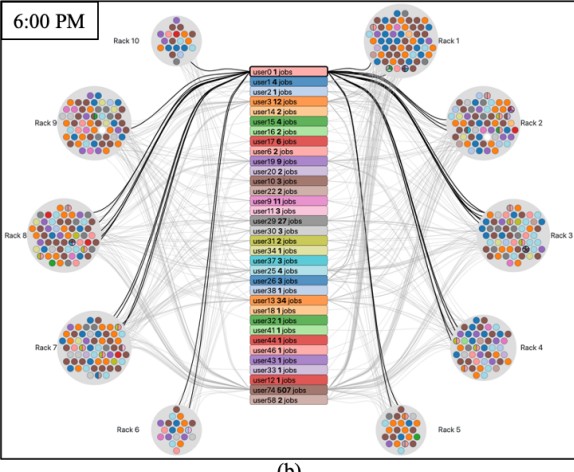

(b)

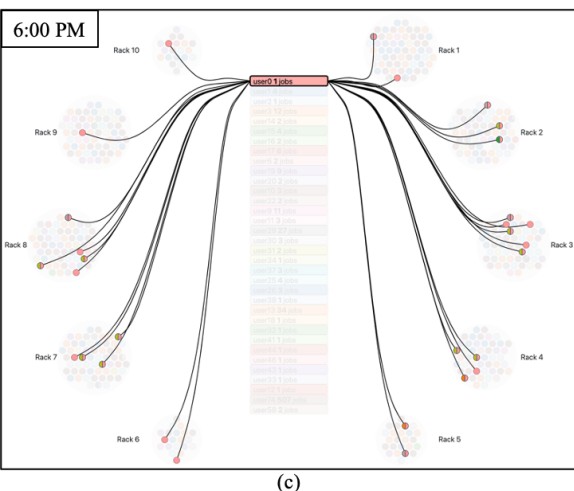

(c)

Figure 2: Snapshots of the main visualization at (a) 5:50 PM and (b) 6:00 PM on 08/14/2020. (c) If we click on *user0*, the highlight of all its links and related compute nodes appears.

## 4.2 Clustering of health states

This use case investigates the health monitoring aspect of the Job-Viewer. As mentioned in section 3, we use color to depict values of a selected health metric from the list of nine. Another option to observe all nine health metrics in a single view is to display each compute node by a radar chart. The radar charts can illustrate all health metrics; however, their size is relatively small for users to recognize quickly. The clustering algorithm can overcome this issue because it clusters all compute nodes to a small number of groups.

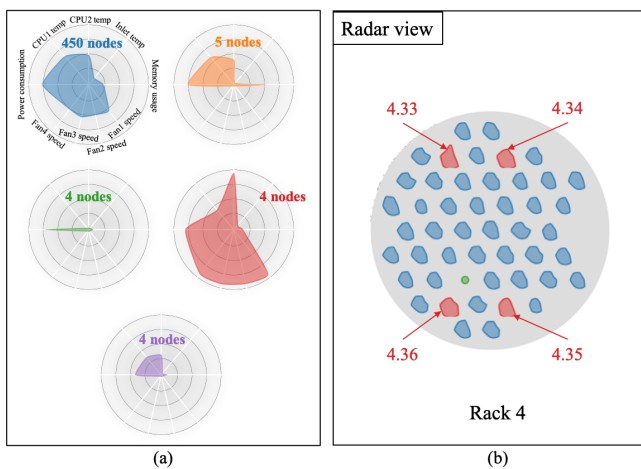

(a)                                     (b)

Figure 3: (a) Result of leader algorithm for all 467 compute nodes on 08/18/2020 at 11:30 AM. (b) Visualization of the compute nodes in rack 4 with radar charts representing the compute nodes. The color of these radar charts depicts its group in the result of the leader algorithm.

Figure 3.a gives the result of the leader algorithm for the system on 08/18/2020 at 11:30 AM. This algorithm clusters 467 compute nodes to 5 groups with different patterns of their health states. The blue group has 450 compute nodes, with medium values of two CPU temperature, four fans' speed, and power consumption. All compute nodes have low inlet temperature, and five of them have high memory usage. We can also see that 13 compute nodes do not have fan speed information, while only four lack information about CPU temperature. The red group has four compute nodes with a common state of high fan speed and CPU2 temperature. Figure 3.b shows these four red compute nodes in rack 4. We have investigated their CPU2 temperature and found that only the compute node 4.33 got heat in its CPU2. Figure 4.b verifies this statement, as the color of compute node 4.33 is red while that of the other three are light green. We can also get the CPU1 temperature of these four compute nodes from figure 4.a. Their CPU1 temperatures are all low due to their corresponding colors. One possible explanation for this event is their location. They may locate near each other, so the three compute nodes (4.34, 4.35, and 4.36) can feel the heat from the compute node 4.33. Then, their fan must work harder to cool the CPUs.

## 4.3 Relation between job and health state

This use case clarifies the relations between jobs and the health states of compute nodes. We firstly look at the time series of CPU2 temperature of the compute node 4.33 in figure 5. The unit of temperature is degree Celcius, and the time takes place in August 2020. The vertical dash line indicates the time step at which we stop the timeline to get the time series. It is 08/18/2020 at 11:30 AM when we investigate the previous use case. The colorful areas highlight periods when a job is running on the compute node. We use text notations, which have similar colors to the corresponding

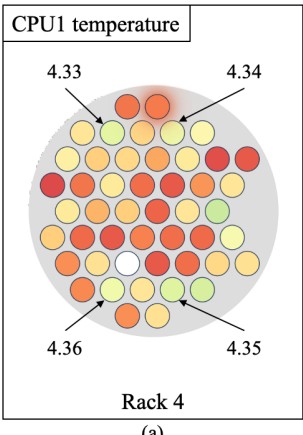  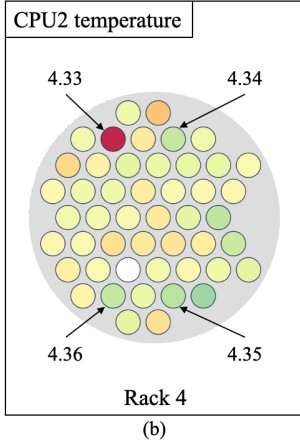

(a)                                     (b)

Figure 4: The visualization of all compute nodes in rack 4. The color indicates (a) CPU1 temperature and (b) CPU2 temperature. Red color means high value, yellow depicts a medium temperature, and green represents low value.

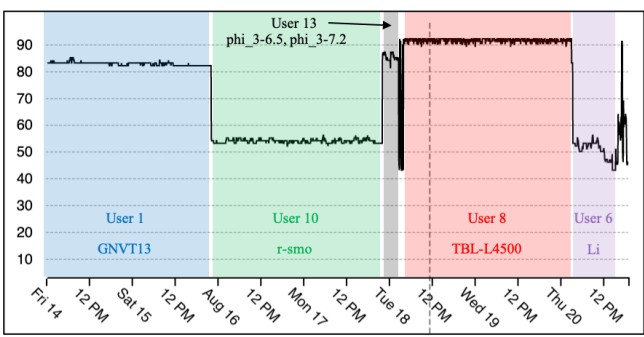

Figure 5: The time series of CPU2 temperature of compute node 4.33. The colorful areas highlight the period when the user with similar color runs his/her job. The vertical dash line indicates the time step at which we take the figure. It is 08/18/2020 at 11:30 AM.

areas, to denote users and their jobs. There are five long jobs on compute node 4.33 over the whole temporal period. None of them overlaps each other. The CPU2 temperature has a high value when *user1* runs his/her job, but the value suddenly reduces when *user10* starts his/her job. The same jump or drop happens when there is a switch of users. Therefore, it is reasonable to state that the CPU2 temperature of compute node 4.33 depends on the job running on it. If we look at the CPU1 temperature of compute node 2.60 in figure 6, we can observe a similar behavior of the relation. Some jobs are responsible for high CPU temperature, while some jobs do not cause hot CPUs.

Can we use these relations to investigate the reason for the irregular hot CPU2 temperature of compute node 4.33, as mentioned in the previous use case? If we compare the user and job running on the compute nodes, namely 4.33 and 2.60, on 08/18/2020 at 11:30 AM, they run only one job of precisely one user. The value of CPU2 temperature of the compute node 4.33 is also higher than other users, such as *user1* and *user13*. This job is suspicious. However, it is impossible to make a strong conclusion about whether this job is the cause of the heat in CPU2 of compute node 4.33 or this compute node has a problem itself. What JobViewer can show to the administrators is the monitoring information. If they want to find the correct reasons for any irregular event, they should do other investigations to see the real causes.

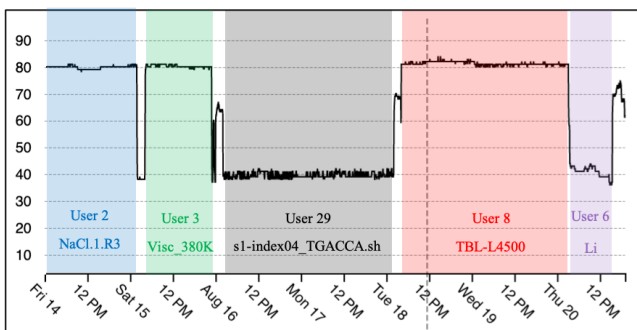

Figure 6: The time series of CPU1 temperature of compute node 2.60. The colorful areas highlight the period when the user with similar color runs his/her job. The vertical dash line indicates the time step at which we take the figure. It is 08/18/2020 at 11:30 AM.

## 5 USER STUDY

### 5.1 Overview

We contact three volunteers, who have experience working with the HPC system (in both academia and industry), and carry out the user study through video calls. The user study begins with an introduction to the JobViewer. The introduction covers all features and functions of four main components. After that, we ask whether the volunteers have questions or any confusion about the application. If they are still not clear about our web-based prototype, we explain carefully again to ensure they fully understand what they can achieve from the JobViewer. The next step is to ask them to answer some questions and record their actions while finding the answers. Finally, we ask whether they have feedbacks on the application or not.

We divide the questions into five tasks as the following:

- **Health metrics**: This task aims to check whether volunteers can gain information about the compute nodes' health states. We require volunteers to select a health metric and name one compute node with a high value of the chosen metric. Also, the volunteers need to point out users linked to that compute node.

- **Job information**: This task checks whether the volunteers know how to get information about a job. We ask them which user's job starts at a particular time step and some compute nodes allocated for the job.

- **Clustering**: This task requires the volunteers to understand how to use the clustering algorithms for detecting the compute nodes with irregular health states. The volunteers need to identify and name all compute nodes with a given pattern of health metrics.

- **Metric vs. Job relation**: This task asks the volunteers to use the time series of a selected metric of a specific compute node to comment on the dependency between the job and the selected metric.

- **General comments**: This task gets the volunteers' feeling when using JobViewer to answer questions in the above tasks. We want to know whether the application is easy for them to find answers to the above questions. Also, we ask whether they think this application is helpful for monitoring activities.

For the task **Clustering** and **Metric vs. Job relation**, we aim to ask the volunteers question related to the use case of compute node 4.33, as mentioned in section 4.2. We hope they can see the benefits of our approach through these questions. One user recognizes some issues with compute node 4.33 and spends time investigating it.

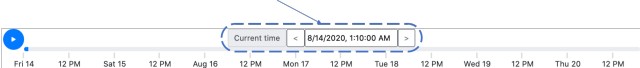

Figure 7: It is difficult to read time information and reach a particular time step in the timeline's old design, so we implement a new component for the timeline. The HPC administrators can click on the right/left button to move toward the time step of interest or type it directly.

### 5.2 Results

Overall, two volunteers can quickly go over the questions and use the application quite correctly, while a volunteer fails almost all the tasks. The first volunteer moves smoothly to all the questions, except reaching a particular time step. It is difficult for him to observe the time on the timeline because it is too small. Also, he takes some issues when trying to reach a specific time step as required by the questions. This volunteer is the only one that spends lots of time on task 4 because he thinks it is interesting to find the reason behind the irregular hot in CPU2 of compute node 4.33, as mentioned in section 4.2. He moves the timeline to look at jobs at different periods and switch to various health metrics to understand the situation. He finally ends up with an assumption about the positions of the four compute nodes. The second volunteer also does well with the tasks, except for the first one. He says that red compute nodes correspond to high values of the selected metric, but he decides to pick up a yellow one to answer the questions. For the task **Metric vs. Job relation**, he replies that the job consuming high CPU usage will cause high CPU temperature. Regarding their opinions about whether JobViewer is helpful for monitoring activities, these two volunteers have common comments. Although JobViewer has a good design and is useful for a human to build up investigations, the monitoring administrators may not spend too much time on any irregular event step by step. What they want is to catch the problems quickly, so they prefer a large monitor with all information and data. About the last volunteer, his only correct answer for the question is to find the node with a high value of *Memory usage*. He comments that the application is hard for him to use because it is challenging to navigate the activities. He also does not understand the use of time series and other stuff.

The first two volunteers also give feedback on how to improve the JobViewer. One is the design of the timeline. Because the whole time interval is long, reaching a particular time step may be a challenging activity. Besides, the text on the timeline may be too small for some application's users to read. It is the reason why we improve our design with a new component above the timeline to make it more useful, as depicted in figure 7. We can directly type the time of interest in this component. Another possible action to get a certain time step is to move near it and use the right/left button to move toward the correct position. Moreover, the second volunteer mentions the scalability of the JobViewer because some HPC clusters may have thousands of compute nodes. Regarding this idea, we believe the graph-based design is suitable for scaling up to a number that is much larger than the current 467 compute nodes. Two reasons that support this argument are as the following.

1. We use color as the primary visual signal to inform the health states of compute nodes. We can select individual HPC users or compute nodes to observe further details and time series. The color helps improve the cognitions if there are too many compute nodes in the system.

2. If we have more rack and compute nodes, we can expend the main visualization because it uses a graph-based design. For example, we can use multiple layers of racks. In this case, the links may look cluttered and crowded. However, it can be overcome by simple highlights.

## 6 DISCUSSION

The strength of JobViewer is its ability to display both system health states and resource allocation information in a single view. It is easy to gain job allocation information in the main visualization, as depicted in section 4.1. The clustering algorithm integrated into the application can quickly show the characteristics of the system health states. From these characteristics, we can point out any compute node with an irregular health state pattern to investigate the problems behind it. Section 4.2 describes a use case for this benefit. Moreover, JobViewer can allow us to observe the relations between jobs and compute node health, as illustrating in section 4.3. This feature highlights jobs and users' behaviors to understand them better for improving or finding suspicious causes of any problem. One volunteer in the user study also finds it interesting to use this feature to investigate the irregular heat in CPU2 of compute node 4.33.

To use JobViewer efficiently, we need the training to know interactions and activities to get desirable information. One volunteer out of three comments on the difficulty of using the application, while the other two can easily go through the tasks. Besides, the timeline's original design is not optimal, so we improve it, as shown in figure 7 and mentioned in section 5.2. Another issue related to JobViewer is that it is not a complete tool for HPC monitoring. We focus on the four design goals rather than an efficient and comprehensive tool for commercial purposes. The JobViewer is an application that can show the advantages of visualization and human-centered computing in a complex task of HPC monitoring.

## 7 CONCLUSION

We have presented an application of human-centered computing in the case of HPC monitoring data. The visualization design bases on the idea of the bipartite graph that has prominence in scalability. The visualization can intuitively show an HPC system with resource allocation information and the system health states. We have demonstrated three use cases of historical data of an HPC cluster with 467 compute nodes to illustrate the proposed approach's usability. Besides, we have carried out a user study with three experienced experts in HPC monitoring. The results point out the strength of JobViewer and its weakness for further improvement in the future.

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
