# OpenReview forum: "A graph-based visualization for monitoring of high-performance computing system"
_graphicsinterface.org/Graphics_Interface/2021/Conference — Submitted to GI 2021_

### Official Review · AnonReviewer2 · 2021-01-12
**Review of "A graph-based visualization for monitoring of high-performance computing system"**

**Rating:** 4
**Confidence:** 5

**Review:**

This paper proposes a visualization system for monitoring and analyzing high-performance computing (HPC) systems. The paper has a clear application domain, and validates the design through three use cases and a user study involving three participants.

The benefits of using visualization and interactive interfaces for understanding and monitoring computer networks (including HPC systems) are generally accepted and acknowledged. The paper describes a number of tools that are designed for this purpose, and discusses the limitations of each tool. It is unclear, however, which of these limitations are being addressed in this paper. In other words, the motivation for this work can be stronger: what specific problems or gaps are being addressed here, given that we already have a number of systems for HPC visualization?

Section 3 goes straight into design goals and descriptions. A detailed task analysis is missing, which is crucial for validating the design goals and requirements.

The bipartite graph visualization is the main visual interface. The paper needs to do a more thorough literature review on related techniques and systems, some of the relevant references include:

Aris, A., & Shneiderman, B. (2007). Designing semantic substrates for visual network exploration. Information Visualization, 6(4), 281-300.

Liu, Z., Lee, B., Kandula, S., & Mahajan, R. (2010, October). Netclinic: Interactive visualization to enhance automated fault diagnosis in enterprise networks. In 2010 IEEE Symposium on Visual Analytics Science and Technology (pp. 131-138). IEEE.

Arendt, D. L., Burtner, R., Best, D. M., Bos, N. D., Gersh, J. R., Piatko, C. D., & Paul, C. L. (2015, October). Ocelot: user-centered design of a decision support visualization for network quarantine. In 2015 IEEE Symposium on Visualization for Cyber Security (VizSec) (pp. 1-8). IEEE.

McLachlan, P., Munzner, T., Koutsofios, E., & North, S. (2008, April). LiveRAC: interactive visual exploration of system management time-series data. In Proceedings of the SIGCHI Conference on Human Factors in Computing Systems (pp. 1483-1492).

I also have some doubts/questions regarding certain visual encoding choices:

- the number of jobs running on a compute node is represented by the thickness of its outline, I think it will be hard for users to see this information.

- If the chosen metric’s value on a compute node varies significantly over two consecutive time steps, the blur effect is used to highlight the sudden change. How exactly is this done?

- An alternative design for the compute node layout is to place them on a coordinate system (like in semantic substrate), as opposed to the circle layout in the current design. This alternate design might reveal more interesting information.

- “If a compute node runs several jobs of multiple users, it has all corresponding colors”: how is this done? Do you divide the node into pies and color each pie? That would be really hard to read.

The effectiveness of the design is evaluated using three use cases and one user study. Without a clear baseline, however, the evaluation feels weak. In particular, I am skeptical about the scalability of the tool. The paper will be stronger if you can convincingly demonstrate how your design outperforms an existing tool through a comparative study.

The writing also needs proofreading, there are many spelling mistakes or grammatical errors:

- In the abstract: “to visualize the monitoring the structural and met- rics data of HPC clusters”

- In the Intro: “TheESE typically complex research instruments ranging from hun- dreds to thorusand of computing nodes”

- “The administrators often consider the compute node health”

---

### Official Review · AnonReviewer1 · 2021-01-13
**Plausible; insufficient validation; needs clarifications**

**Rating:** 6
**Confidence:** 3

**Review:**

The paper proposes a user interface with a novel visualization to monitor a HPC cluster, health of its nodes, and user/job allocation. The proposed solution visualizes the system as a bipartite graph, where job are connected with nodes via graph edges, clearly indicating the correspondence, and nodes are colored based on the desired factor, e.g. CPU temperature. The paper validates its design via a small-scale user study, showing 2/3 volunteers succeeded in using hte system.

The designed interface seems plausible to me, the overall idea of the paper, as well as the design goals and outlines in Sec.3, make sense. I have a few complaints, however: A few design decisions, however, seem to be poorly justified, e.g. the use of radar charts to visualize all health metrics at the same time. Can humans actually quickly understand those? What do they add?
Also, I was not very clear on the design of the user study: were the five tasks developed with an expert? Do they reflect typical tasks of a system administrator, regardless of the UI, or were they created specifically to showcase the functionality of this interface. I would love a clarification on that (in the former case it is a more interesting usability study). I am also not sure how conclusive the user study is when two out of three succeeded to use, and one failed. I````````'d perhaps suggest testing on more volunteers?

Finally, the text should be edited and spell-checked by a native English speaker, it has numerous errors, at times completely unclear, or style issues ('thick of its outline', 'irregular hot', 'and other stuff').

Overall, I think it is a reasonable approach, but I would like to see the edits and corrections above.

---

### Official Review · AnonReviewer3 · 2021-01-14
**Interesting paper on monitoring high performance computing systems**

**Rating:** 5
**Confidence:** 3

**Review:**

This paper presents Jobviewer, a visualization that helps monitoring a high perform computing system. The visualization uses a sequence of rectangles to show jobs along with curved links to connect to racks. Using this visualization the user can monitor the health of a computer node. The authors ran an informal user study with three participants to understand the utility of the design.

The visualization seems to have some useful components, however, the design lacks justification of key design decisions. The paper could be strengthen by explaining how the discussions with users had lead to the design. Another issue is that the visualization is not also very novel. There are similar visualizations that exist although for different data (e.g. [1]).

The user study of the paper is very limited with three participants who just provided some informal feedback which is not sufficient to validate the system. As a minor suggestion for writing, it would be helpful if the user study section is written in past tense (since they have already participated in the study).

Overall, this is an interesting work however due to lack of novelty of the visualization and design rationale it is difficult to accept the paper in the current form.

---

### Meta-Review · Area_Chair1 · 2021-01-14

**Recommendation:** Reject
**Confidence:** 3

**Metareview:**

The reviewers think that key design decisions lack justification and alternative designs may be better. The paper could be stronger by validating the design choices through explicit discussions of alternatives and rationales.

The user study is also considered weak given that only three participants were involved, and there was no baseline for comparison. Finally, the writing needs editing and proofreading by a native speaker.

---

### Decision · Program_Chairs · 2021-01-16

Reject